# A Short Course of Standard Velcade/Dexamethasone Followed by Unlimited Weekly Maintenance Therapy Is an Effective Treatment in Relapsed/Refractory Multiple Myeloma

**DOI:** 10.3390/cancers16223805

**Published:** 2024-11-12

**Authors:** Harini Acharya Gangur, Harsha Trivedi, UshaSree Chamarthy, Anas Al-Janadi, Gordan Srkalovic

**Affiliations:** 1School of Medicine, Lake Erie College of Osteopathic Medicine—Bradenton, Bradenton, FL 34211, USA; hgangur96773@med.lecom.edu; 2Herbert-Herman Cancer Center, University of Michigan Health-Sparrow, Lansing, MI 48912, USA; harsha.trivedi@umhsparrow.org; 3Sparrow Regional Cancer Center Lansing, Sparrow Health System, Lansing, MI 48912, USA; ushasree.chamarthy@gmail.com; 4Department of Medicine, College of Human Medicine, Michigan State University, East Lansing, MI 48823, USA; anas.al-janadi@corewellhealth.org

**Keywords:** multiple myeloma, relapsed/refractory, bortezomib, Velcade, clinical trials, shortened therapy

## Abstract

Multiple myeloma is the second most common hematological malignancy and remains incurable with current therapies. Here, we show that instead of the standard eight cycles of treatment with Velcade/Dexamethasone, the abbreviated four cycles of Velcade/Dexamethasone, followed by unlimited maintenance with Velcade, is effective and well tolerated. This may represent an alternative to other established treatments, with potential financial benefits.

## 1. Introduction

Multiple Myeloma (MM) accounts for about 10% of all hematologic malignancies and is characterized by a relapsing disease course [1]. Despite significant improvements in patient outcomes following the introduction of immunomodulatory drugs (IMiDs) and proteasome inhibitors (PIs) in the first-line setting, and improved survival in the last fifteen years, most patients eventually relapse [2]. Thus, the management of relapsed and/or refractory MM (RRMM) remains a challenge [3,4].

With the increasing recognition of the inherent clonal heterogeneity and genomic instability of the plasma cells influencing both inherent and acquired therapeutic resistance, it is becoming increasing complex and difficult for treating physicians to choose the best option for a patient in terms of the efficacy, safety and sequence of the available therapies for treatment [5]. Due to the many newer drugs available for treating MM, it has turned into a chronic disease with a series of remissions and relapses. In the relapsed setting, it is hard for physicians to choose the most appropriate agent to use. The treatment choice is additionally complicated by patients and disease-related factors such as age, cytogenetics and molecular profile, pre-existing comorbidities, pace of relapse, and previous therapies, including response and residual toxicity [4,6,7]. Thus, with each relapse, the ultimate goal is to find the best therapy that can prolong the remission period and overall survival of the patient [8,9].

In the past, due to thalidomide (Thal) toxicity, combinations of the IMiD agent lenalidomide with dexamethasone (RD) and the PI Velcade (bortezomib) with dexamethasone (VD) have become the standard of care in RRMM. This was based on large, randomized studies versus dexamethasone (Dex) alone [10,11,12,13,14]. In previous studies with bortezomib, this agent was given intravenously (IV) with a significant incidence of mostly sensory peripheral neuropathy, with grade 3 or higher events ranging between 8 and 13% [10,13,14].

In 2011, a randomized, phase 3, non-inferiority study was published, showing that subcutaneous (SC) bortezomib was non-inferior to IV, with an improved safety profile [15]. In this study, the incidence of peripheral neuropathy grade 3 or higher was 6% for SC vs. 12% for IV bortezomib. SC bortezomib was approved in the US and EU in 2012 and became the standard of care.

Many newer agents, like second-generation IMIDS, anti-CD-38 antibodies, e.g., daratumumab, isatuximab, bispecific antibodies, and CAR-T cell therapies, have improved the morbidity and mortality outcomes of this disease [16,17,18,19,20,21,22,23,24,25].

However, cost and tolerance are becoming more of a concern and can direct treatment toward “older” treatments, which are “tried and true”.

We hypothesized that abbreviated induction and unlimited maintenance treatment with Velcade will be inferior to standard induction of eight cycles and unlimited maintenance with Velcade until toxicity or progression. This phase II study was designed to test that hypothesis.

## 2. Materials and Methods

### 2.1. Patient Selection

Patients were recruited for this investigator-initiated trial from 6 June 2006 to 24 September 2013 at three medical centers in the US: Western Maryland Health System, Michigan State University, and Sparrow Hospital System. The study was permanently closed in 2017. At the time of closure, 24 patients were enrolled out of the targeted goal of 30 patients. At that time, all the patients treated with maintenance progressed. Patients who achieved a complete response (CR) were observed. The eligibility requirements included men and women with a confirmed diagnosis of previously treated, active MM. Patients in the trial must have relapsed and/or refractory disease, as defined by the Bladé criteria [26]. Patients must have recovered from all prior toxicities and signed the informed consent approved by the Sparrow Hospital and Michigan State Institutional Research Review Committees (IRRCs).

The inclusion criteria included patients with RRMM and progression of disease after the last therapy. The eligibility criteria also included ECOG PS < 3, life expectancy > 3 months, use of an acceptable method of contraception (where applicable), and lab values: platelets (Plt) ≥ 50 × 109/L or ≥ 30 × 109/L if extensive bone marrow infiltration, hemoglobin (Hgb) ≥ 7.5 g/dL, absolute neutrophil count (ANC) ≥ 0.75 × 109/L, corrected serum calcium < 14 mg/dL, aspartate amino transferase (AST) < 2.5 × ULN, alanine transaminase (ALT) < 2.5 ULN, and creatinine clearance ≥ 30 mL/min 14 days before study drug administration. The serum and urine protein electrophoresis and immunofixation (IFE), bone marrow aspirate and biopsy, skeletal survey, β-2 microglobulin, and C reactive protein (CRP) were collected to assess the efficacy of the treatment.

The exclusion criteria included chemotherapy within 4 weeks or radiotherapy within 3 weeks prior to enrollment, corticosteroids (>10 mg/day of prednisone or equivalent) within 3 weeks of enrollment, immunotherapy within 8 weeks, plasmapheresis within 4 weeks, or major surgery within 4 weeks, allergy to mannitol, grade 2 or more peripheral neuropathy as per the NCI common toxicity criteria (CTC), any major cardiac disease, other uncontrolled comorbidities, active or history of hepatitis or HIV infection or ongoing other infections, or women who were pregnant or breast feeding, and patients on other clinical trials were excluded from the trial.

### 2.2. Study Design and Treatment

Patients in the study received 4 standard induction cycles and were followed by maintenance of once-weekly cycles (5 weeks on, 1 week off) until progressive disease (PD), intolerance, or investigator/patient decision to stop therapy. The standard cycles were as follows: Velcade 1.3 mg/m^2^ IVP or SC (per patient choice) was administered on days 1, 4, 8, and 11, followed by a 10-day rest period, Q21 days × 4 cycles; followed by maintenance therapy with Velcade 1.6 mg/m^2^ (or previously tolerated dose) IVP/SC on d1, 8, 15, and 22, followed by a 13-day rest period and repeated Q36 days until progressive disease (PD), intolerance or investigator/patient decision to stop therapy. Dexamethasone was given at 20 mg PO on the day of the Velcade injection and 20 mg PO the day after. Patients who experienced a complete response (CR) at cycles 2 to 4 of induction continued receiving Velcade and dexamethasone on the standard regimen for 2 cycles beyond CR. Patients who achieve CR on the weekly schedule received an additional two cycles beyond confirmation of CR. These patients were then observed during therapy. Dose escalation was not allowed in any patient.

The toxicities were assessed according to the NCI CTC, version 3.0. For patients experiencing toxicities of grade 3 or 4, the Velcade dose was held. As per the judgement of relatedness to Velcade, treatment was held if there was grade 4 hematologic toxicity or greater than grade 3 non-hematologic toxicity until it resolved. For other toxicities, it was held for 2 weeks until it resolved to grade 1 or better. If, after Velcade had been held and the toxicity did not resolve, then the study drug was discontinued. If the toxicity resolved, then Velcade was restarted at the same dose level if only one dose was held.

If two doses were held (either consecutive or two out of one cycle), then the Velcade dose was reduced to the next level (1.3; 1 and 0.7 mg/m^2^). If the last tolerated dose was 0.7 mg/m^2^ and the patient needed a dose reduction, Velcade was discontinued. For patients who required a delay of a dose due to toxicity, the dose was omitted. The FACT neurotoxicity-directed questionnaire was used as a tool for determining the presence and intensity of neuropathic pain and/or peripheral neuropathy from the patient’s perspective. Neurotoxicity was also assessed by both nurses and physicians separately at the beginning of each cycle.

### 2.3. Response and Safety Assessment and Criteria

Standard response definitions were used to determine response and progression based on the Bladé criteria [26]. Measurable disease was defined by the presence of quantifiable protein criteria. Acceptable protein criteria were detectable levels of serum M protein, quantified by serum protein electrophoresis (SPEP), and/or urine M protein (Bence–Jones Protein), and 24 h urine protein electrophoresis (UPEP). Patients who had a serum free light chain (FLC) at baseline or those with FLC-only disease were followed by serum FLC assay. Response or progression was confirmed by a second disease assessment (after 6 weeks) prior to the initiation of any new therapy. A skeletal survey was conducted at baseline and then was not required for the assessment of response unless clinically indicated.

### 2.4. Subject Evaluation

The patient evaluations included medical history, including current medications and therapies, physical exam, echocardiogram, chest X-ray, vital signs, and clinical laboratory studies, complete blood count (CBC), comprehensive metabolic panel (CMP), urinalysis (UA), and amylase before every cycle of treatment. A pregnancy test and HIV serology were performed at baseline. All the baseline evaluations were performed prior to beginning the study therapy. The weight and performance status were assessed every 3 weeks for the first four cycles of induction and then every time at the beginning of the maintenance cycles (q 36 days) until progression. Patients were monitored for toxicity twice weekly during the first four cycles of treatment, then weekly prior to each maintenance cycle or at more frequent intervals at the investigator’s discretion.

### 2.5. Statistical Analysis

All the analyses were performed using R and SAS statistical software. (SAS version 9.4 and R studio 2 April 2024 build 764) The confidence intervals for the response rates were calculated using the binomial exact method and the methods of Kaplan and Meier were utilized to evaluate the survival outcomes [27]. The plan was to enroll 30 patients. This corresponds to an observed response rate of at least 17%. This treatment was considered ineffective if fewer than five responses observed. Data were analyzed to determine the induction response rate (IRR) at the end. Data were also analyzed to determine the progression-free survival (PFS) and overall survival (OS). Univariate analysis was performed. Multivariate analysis was not performed as only one variable was statistically significant.

## 3. Results

### 3.1. Study Population Characteristics

Twenty-four patients were treated between June 2006 and September 2013. The median age was 67 years. The previous therapies received by these patients were as follows: seven patients received thalidomide, two patients received thalidomide in combination with lenalidomide, nine patients were treated with Revlimid with dexamethasone, out of which one patient was previously treated on the S0777 trial, one patient had an autologous bone marrow transplant, two patients received radiation to plasmacytomas followed by autologous stem cell transplant, and three patients had received melphalan plus prednisone in the first line. Twenty patients were tested for cytogenetics, and three patients had apparent abnormalities. There was one patient each with p21 deletion, p13 deletion, and translocation q21 with p11 deletion. There were no high-risk cytogenetics patients (Del 17p, t 4:14 or t 14:16) identified via metaphase cytogenetics. Of the 24 patients analyzed, 9 (38%) were originally ISS stage I, 9 (38%) stage II, and 6 (25%) stage III (Table 1).

### 3.2. Efficacy

The response analysis included the 24 patients who were treated. Among these patients, the best overall response observed was a complete response (CR) for six (25%) patients, a partial response (PR) for eight patients (33%) and stable disease (SD) for six patients (25%). Thus, 14 of the 24 patients (58.3%, 90% CI: 39.7, 75.4) had a PR or better response. Four patients (17%) had progressive disease (PD) during induction (first four cycles). After four cycles of induction were completed, four patients had a complete response, eight patients had a partial response, seven had stable disease and five patients had PD. In total, 12 of the 24 (50%, 90% CI: 31.9, 68.1) patients experienced an induction response of PR or better.

After maintenance therapy was completed, there were two additional responses, making a total of 14 (58.3%) patients who had a response on the trial. The quality of the response improved on maintenance therapy and one PR and one SD converted into CR. The median OS was 26.9 months (95% CI: 16.37, 48.1) (Figure 1) and the median PFS was 11.3 months (95% CI: 8.51, 44.68) (Figure 2). Moreover, 16 out of 24 patients had measurable serum free light chains (sFLCs).

Univariate analysis was performed for different variables in the study (Table 2). Only one variable, ISS stage III, was found to be statistically significant for both PFS (*p* = 0.002; 95% confidence interval (1.68, 15.8) HR 5.16) and OS (*p* = 0.001; HR 7.06, 95% CI (1.94–25.7%), Multivariate analysis was not performed since only one variable was statistically significant. This result, even though pointed, shows the promise of further studies in stage III to confirm these findings.

### 3.3. Safety Overview

The treatment duration ranged from one month to 6.4 years. Treatment was well tolerated. The mean number of cycles of treatments was 9.6 (95% CI, 4.81, 14, 34). The median number was 5.5 cycles. The most common grade 1 toxicity was sensory neuropathy (25%), followed by arthralgia (17%). The grade 2 toxicities included fatigue (30%), followed by sensory neuropathies (20%), arthralgia (21%), and motor neuropathies (12%). The grade 3 toxicities included fatigue (58%), sensory neuropathy (54%), thrombocytopenia (50%), motor neuropathy (33.3%), diarrhea (20%), arthralgia (8%), hypotension (4.2%), vasovagal episodes (4.2%), blurred vision (4.2%), headache (4.2%), and edema in limbs (4.2%). The grade 4 toxicities reported were thrombocytopenia (12.5%), fatigue (12.5%), sensory neuropathy (12.5%), and motor neuropathies (8.3%) (Table 3).

The most common serious AEs were non-protocol related (accidents, comorbidities leading to death, etc.). None of the patients died due to treatment-related toxicities. A total of 18 patients discontinued the study protocol: 11 patients due to progressive disease, 4 voluntary withdrawals, and 3 patients due to accident and comorbidities. One patient transformed to plasma cell leukemia. Twelve patients (50%) had at least one dose reduction per the study protocol guidelines.

The toxicities leading to dose reductions were sensory neuropathy (30%), diarrhea (20%), and motor neuropathy (5%). Two patients started on a reduced dose of Velcade (1.0 mg/sq m) at cycle one of induction due to ongoing kidney problems (increased serum creatinine at baseline). The most common tolerable induction dose was 1.0 mg/m^2^ in patients aged 65 or older. Only one patient had to be dose reduced for maintenance treatment with a Velcade dose of 0.7 mg/m^2^ due to neuropathy. The rest of the study patients tolerated higher maintenance doses without major problems.

## 4. Discussion

Patients with MM who have received multiple prior therapies are a challenge to treat. The progression outcomes become progressively worse with an increasing number of prior therapies [9,16,28]. Therefore, effective and tolerable treatments must be available for this heterogeneous patient population.

Currently, therapies like daratumumab-based, ixazomib, and elotuzumab combinations and various pomalidomide-based regimens have shown efficacy [19,20,21,22,23,24,25].

For aggressive relapses, anthracycline-containing regimens may be useful [29]. Venetoclax has been shown to be effective in the t (11;14) subtype of MM [30].

Two of the most exciting newer investigational options are chimeric antigen receptor T cells (CAR-T) and bispecific antibodies (BITES) [22,24]. To date, three BiTES therapies are approved for the treatment of RRMM. Both these therapies come with a high cost and their own set of side effects of cytokine release syndromes and immune effector cell associated neurotoxicity syndrome (ICANS) and a higher infection risk [31,32]. Another exciting development of ongoing research in the RRMM area is angiogenesis. Blocking angiogenesis with small molecules like DARPins in pre-clinical studies has been shown to potentiate the effect of bortezomib, thus decreasing the rate of progression. Such novel approaches are necessary to investigate further in clinical trials [33].

Currently, three or four drug therapies are given in the USA to better control the progression of disease. This has improved outcomes. We cannot say which combination is the best as larger randomized trials with comparative arms are still pending [34].

In a scenario where these newer agents are not accessible, this is a viable option. The NCCN lists the use of Velcade/Dex as a category 1 treatment in certain circumstances; for example, after two prior therapies, including IMID and a proteasome inhibitor and with disease progression on/within 60 days of completion of last therapy—NCCN guidelines Index version 4.2024, pages 4/5 [35]. Our study indicates that the use of Velcade/Dex in the protocol described in this paper is an effective and tolerable treatment for patients in the US as well as in other countries where the availability of the three or four drug regimens is limited by the cost as well as in patients whose tolerability is restricted due to side effects.

The objective of this trial was to investigate the activity and tolerance of abbreviated induction with Velcade/dexamethasone as well as the tolerability of extended maintenance with the same treatment. The targeted recruitment was 30 patients, but only 24 patients were enrolled and analyzed. The study recruited patients from community hospitals and not academic centers. This was most probably a limiting factor. Due to the slow recruitment, the study was closed early after a statistician advised that with a reduced number of patients (*n* = 24), if the response is 32%, then it will have the same power (80%) for analysis (personal communication). The response rate was found to be 58.3% in the statistical analysis.

Our study showed that out of 24 evaluable patients, 14 patients (58.3%) had a PR or better response. Treatment was well tolerated by most patients. The median PFS was 13.82 months and the median OS was 29.08 months. These responses are comparable to more intense (biweekly) treatment with Velcade. We thus rejected our hypothesis that this design of abbreviated induction cycles and unlimited maintenance treatments in RRMM will be inferior to standard treatment.

The response rates in patients with RRMM vary widely. New, less toxic, and more effective IMIDs showed significant activity in RRMM. Lenalidomide, as a single agent, showed an ORR of 26% in patients treated previously with three prior lines of therapy [14]. In combination with dexamethasone, in patients after more than two previous lines of therapy, lenalidomide showed an ORR of roughly 60% [11,12]. Combinations with chemotherapy, proteasome inhibitors, and monoclonal antibodies produced an ORR from 64–92% in several studies [5]. Single agent bortezomib had an ORR 27–41%, with a better VGPR of 7–19% [10,13,26,30]. The responses in these studies depended very heavily on the number of previous treatment lines [30]. In combination with other agents, bortezomib had an ORR 34–87%, with a better VGPR between 11 and 56% [5].

When compared to historical results, the ORR in our study is in line with others. A total of 14/24 (58.3% CI 35.7, 75.4) patients had ORR (PR + CR), and although this is a small study, the CR seems slightly better than the VGPR in other studies (5–28%) (5). Similarly, the PFS and OS in our study reflect the results from larger studies when single agents are combined with dexamethasone [5]. These results signify a promising indication to pursue larger studies with increased numbers of patients. Moreover, the univariate analysis (Table 3) also points out that this is more effective in patients with stage III disease (*p* < 0.002). It is indeed a pointed conclusion since it is a univariate analysis, and it needs to be further validated in a larger study. Nevertheless, it further indicates that this mode of treatment needs further studies.

More importantly, an abbreviated course of twice-weekly Velcade/dexamethasone (four cycles as opposed to eight standard ones) did not reduce the efficacy of the treatment and seems to be increasing compliance and reducing toxicity. In addition, during the maintenance phase, two patients had an improved response from PR and SD to CR with no added toxicity or toxicity-related withdrawal from the study. One patient stayed on the maintenance for 6.3 years without signs of progressive side effects. Another patient was treated for almost 3 years (2.8) and two more for close to 2 years (1.9 and 1.7 years). All of them discontinued the study treatment due to progressive disease, not toxicities.

Our patients belonged to a relatively good prognostic group. Most of them (22/24) received only one prior line of therapy. The majority (18/24) were stage I and II. There were no patients with high-risk cytogenetics. The median age of the patients was as expected for MM patients (67 years). The patients were mainly of Caucasian ethnicity as the hospitals where it was opened mainly served this population. The most common grade 3 and 4 adverse events were fatigue (unrelated) and low platelets as well as neuropathies. This is a common finding across most of the IMID and proteasome inhibitor therapies.

Maintenance with Velcade, as opposed to lenalidomide, is not considered a standard approach, particularly post-autologous stem cell transplant (ASCT). This is mostly due to robust data showing lenalidomide’s efficacy post-ASCT [36,37]. A meta-analysis involving 1209 patients within three randomized studies demonstrated a significantly prolonged OS when compared to controls [38,39]. However, the increased incidence of second primary malignancies after lenalidomide maintenance must be part of any serious physician–patient discussion. That is also the main reason why the optimal duration of maintenance therapy with lenalidomide is currently not established [34].

Velcade maintenance in the post-ASCT period in newly diagnosed MM patients was investigated in the HOVON-65/GMMG-HD4 and Spanish Myeloma studies. The first studies showed that Velcade-based induction (as a part of PAD) and single-agent Velcade maintenance significantly improved the PFS and OS in all patients, including those with high-risk cytogenetics (del 17 p) and patients with renal failure [40,41]. The Spanish study showed superior PFS, but not OS, with Velcade maintenance in combination with thalidomide when compared to single-agent thalidomide or interferon-alpha 2b [41].

RRMM Velcade in combination with dexamethasone was shown to be tolerable even in unfit or frail patients [42]. For high-risk patients, bortezomib-based maintenance should be considered [1]. In this study, the maintenance of Velcade dose was 1.3 mg/m^2^ given every 2 weeks for 16 treatments. Maintenance therapy improved the quality of the response and converted previously PRs into one CR and four VGPRs and produced a decrease of M protein in 11/40 patients [43].

In our study, we found very good tolerance of the maintenance treatment and the possibility of a long maintenance duration without the worsening of side effects, particularly peripheral neuropathy. The starting dose for maintenance in our study was 1.6 mg/m^2^, but due to side effects, all the patients started at lower doses. We also observed a response improvement after maintenance. Two patients had an improved response from PR and SD to CR and four VGPRs and produced a decrease of M protein in 11/40 patients [43]. This does show that this approach is feasible and indeed reduces exposure to the drug as well as the cost of treatment.

The limitations of this study included sex (males are more represented than females) and race, as this study included a mostly Caucasian population. Also, it had a single arm study design, and the sample size was small, with 24 patients recruited out of the targeted 30 patients. Further studies must be conducted taking these factors into account. It was an investigator-initiated clinical trial opened at community cancer centers with limited resources both financially and personnel-wise. Recruitment of patients was a challenge due to RRMM patients going to academic centers, limited financial support and patient transportation difficulties for research visits. This is a reality of community practices. However, the participant number (*n* = 24) was statistically powered enough to show significance in terms of the efficacy and safety of abridged treatment in induction followed by unlimited maintenance treatment.

This opens the path for a potential larger study with a control group, more diverse patients and potentially a comparative arm with this design. This will further help to strengthen these findings and make them more robust. The shortened duration also provides financial benefits to many countries struggling with healthcare costs.

## 5. Conclusions

In conclusion, our study showed that a short course of Velcade/dexamethasone, followed by indefinite maintenance with Velcade, is a feasible and well-tolerated treatment in RRMM patients. The response rates, PFS and OS are in line with larger studies using a standard combination of proteasome inhibitors and steroids. We showed that an abbreviated initial course (four instead eight cycles) of Velcade/dexamethasone produced a very strong response rate and maintenance treatment can be applied for a very long period in individual patients without major complications and the loss of treatment activity. Despite the introduction of new agents into the RRMM arena, long-term maintenance with Velcade/dexamethasone could potentially become an additional attractive alternative to new agents due to the good tolerance and potential financial benefit in many parts of the world. The results of this study can be further tested in a larger clinical trial with controls and a comparative arm.

## Figures and Tables

**Figure 1 cancers-16-03805-f001:**
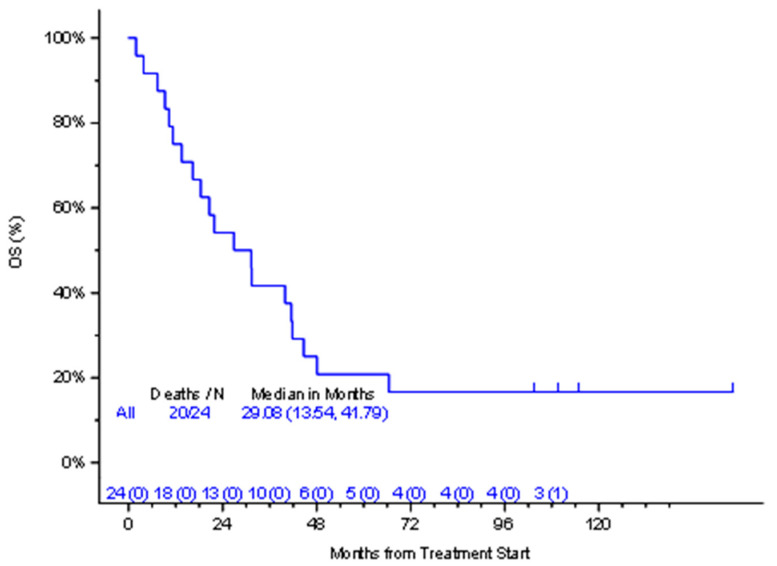
Overall survival (OS) with number at risk.

**Figure 2 cancers-16-03805-f002:**
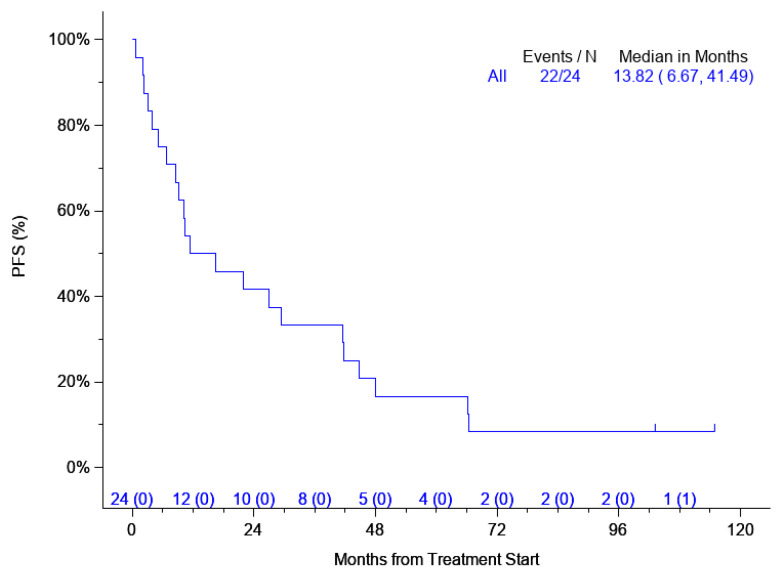
Progression-free survival (PFS) with number at risk.

**Table 1 cancers-16-03805-t001:** Baseline characteristics of patients.

Characteristic	*N* (%)
Sex	
Male	16 (67)
Female	8 (33)
Race	
Caucasian, non-Hispanic	22 (92)
African American	2 (9)
Lines of Treatment	
Previous lines of treatment—One line	22
Number of prior lines of treatment—Two lines	2
Staging	
Stage I	9 (38)
Stage II	9 (38)
Stage III	6 (25)
Cytogenetics	
Apparent abnormalities	3 (6)
Abnormalities detected	P21 deletion, p13 deletion, t q21 and p11 deletion.
Total patients tested	20 (83)
Median age (range)	
Age (years)	67 (51–85)

**Table 2 cancers-16-03805-t002:** Univariate analysis (note that there is no multivariate analysis since only one variable was statistically significant).

			PFS		OS	
	Variable	*n*/*N* (%)	HR (95% CI)	*p*-Value	HR (95% CI)	*p*-Value
Univariate	Age ≥ 65 year	14/24 (58%)	1.35 (0.56, 3.21)	0.502	1.29 (0.52, 3.17)	0.579
	Female	8/24 (33%)	1.37 (0.55, 3.38)	0.499	1.71 (0.68, 4.33)	0.250
	b2m ≥ 3.5 mg/L	10/24 (42%)	1.61 (0.68, 3.81)	0.273	2.12 (0.87, 5.15)	0.090
	Albumin < 3.5 g/dL	10/24 (42%)	1.00 (0.43, 2.35)	0.997	1.00 (0.41, 2.44)	0.997
	sLFC ratio (involved/uninvolved) ≥ 100	12/23 (52%)	1.61 (0.67, 3.84)	0.282	1.06 (0.43, 2.62)	0.896
	ISS stage I	9/24 (38%)	0.72 (0.30, 1.74)	0.464	0.52 (0.20, 1.36)	0.174
	ISS stage II	9/24 (38%)	0.64 (0.27, 1.53)	0.311	0.78 (0.32, 1.92)	0.592
	ISS stage III	6/24 (25%)	5.16 (1.68, 15.83)	0.002	7.06 (1.94, 25.75)	<0.001

HR, hazard ratio, 95% CI, 95% confidence interval, *p*-value from the score Chi-square test in the Cox Regression NS2, multivariate results not statistically significant at the 0.05 level. All univariate *p*-values reported regardless of significance. Multivariate model uses stepwise selection with entry level 0.1 and variable remains if meets the 0.05 level. A multivariate *p*-value greater than 0.05 indicates variable forced into model, with significant variables chosen using stepwise selection.

**Table 3 cancers-16-03805-t003:** Number of patients with a given type and grade of adverse event per CTCAE v3.0.

Adverse Event	Grade 3 (*N*)	Grade 3 (%)	Grade 4 (*N*)	Grade 4 (%)
Fatigue	14	58	3	12.5
Platelets	12	50	3	12.5
Diarrhea	5	20.8	0	
Motor neuropathy	8	33.3	2	8.3
Sensory neuropathy	13	54	3	12.5
Arthralgia (general)	2	8.3	0	
Hypotension	1	4.2	0	
Vasovagal episode	1	4.2	0	
Blurred vision	1	4.2	0	
Pain–headaches	1	4.2	0	
Edema in limbs	1	4.2	0	

## Data Availability

Data are not available on public platforms due to HIPPA restrictions. However, data are presented blinded in this paper in form of tables, figures and text.

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
