# Peer review of "A Short Course of Standard Velcade/Dexamethasone Followed by Unlimited Weekly Maintenance Therapy Is an Effective Treatment in Relapsed/Refractory Multiple Myeloma"

_cancers, 2024, doi:10.3390/cancers16223805_

Round 1

Reviewer 1 Report (Previous Reviewer 1)

Comments and Suggestions for Authors

Improved.

This manuscript is a resubmission of an earlier submission. The following is a list of the peer review reports and author responses from that submission.

Round 1

Reviewer 1 Report

Comments and Suggestions for Authors

The study in 24 patients within a 7 year recruitment period confirms that a 2 drug regimen after limited cycles can be - due to various resons- be further reduced, however and very unfortunately, it seems unlikly that this paper finds many readers.

It is also advised that the authors

1. Name exactly the prior regimens before Vd as the 2. line treatment option

2. Give „numbers at risk“ below all Kaplan Meier curves for consistency .

3. Perform and show uni- and multivariate analysis of those pts profiting from this Vd approach.

4. Show both PFS and OS side by side,

the cut-off Kaplan Meier curves with < vs> 100 LC values do not add to the paper and should be removed.

5. The discussion is by far too long and can be reduced to 1/3 of its current length,

moreover and most importantly, the data should not be exaggerated in its meaning.

6. Since 3 and 4 agent combos are typically given today and Cd38ab-VRd is one standard for both TE and NTE pts, one wonders, where for 2. line treatment: Vd should be used and this from an US center, where all possible options are available.

A table or even better a graph, that illustrates the significance of such a therapy option would add to shed some light on the authors intention, „that is is really the right way to go in which pts“.

8. The authors need to emphasize the criticisms of their study: a) very limited pts, b) long recruitment period, c) what percentage of eligible RRMM pts received other options, most likely >95%, d) the one institution/fes institutions involving ? Recruitment, e) no confirmation phase 3 study planned or done, etc…

Reviewer 2 Report

Comments and Suggestions for Authors

The authors conducted an open-label, Phase II clinical trial to evaluate the efficacy and tolerability of a modified treatment approach using Bortezomib (B) and dexamethasone (D) in relapsed/refractory multiple myeloma (RRMM) patients. The standard treatment protocol for RRMM includes 8 cycles of intravenous push (IVP) injections of B and oral D, which can increase toxicity.

In this trial, RRMM patients who had received at least one previous therapy were eligible for participation. The patients received B 1.3 mg/m2 IVP or subcutaneous (SC) injections on days 1, 4, 8, and 11, followed by a 10-day rest period, repeated every 21 days for 4 cycles. After the induction phase, patients entered a maintenance phase with once-weekly B 1.6 mg/m2 IVP or SC injections on days 1, 8, 15, and 22, followed by a 13-day rest period, repeated every 36 days. Additionally, patients received D 20 mg on the days of B administration and for 28 days after.

Out of the 24 enrolled patients, 6 (25%) achieved a complete response (CR), 8 (33%) had a partial response (PR), and 9 (37.5%) had stable disease (SD). Overall, 14 out of 24 patients (58.3%) achieved a PR or better response. Four patients experienced progressive disease (PD) during the induction phase. The most common grade 3 toxicities observed were fatigue (58%), sensory neuropathy (54%), and thrombocytopenia (50%). Grade 4 toxicities included thrombocytopenia (12.5%), fatigue (12.5%), and sensory neuropathy (12.5%).

The authors concluded that the short course of B and D followed by maintenance therapy with B was well-tolerated in RRMM patients. The long-term maintenance approach using B and D could potentially serve as an alternative treatment option for RRMM patients, possibly replacing the need for new agents.

Limitations:

  1. Small Sample Size: The trial had a small sample size, with only 24 patients accrued out of the targeted goal of 30 patients. This limited sample size may affect the generalizability and statistical power of the study's findings.

  2. Single-Arm Study Design: The study used a single-arm design, meaning there was no control group for comparison. This design limits the ability to draw definitive conclusions about the effectiveness of the treatment and assess its superiority over other existing treatments.

  3. Incomplete Target Enrollment: The study did not reach its targeted enrollment of 30 patients. The incomplete enrollment may introduce selection bias and affect the representation of the patient population under study.

  4. Limited Diversity: The study was conducted at three medical centers in the US and had a predominantly Caucasian, non-Hispanic patient population. This lack of diversity may limit the generalizability of the findings to more diverse populations.

  5. Retrospective Data: The data collection period spanned from June 2006 to September 2013, and the study was permanently closed in 2017. The use of retrospective data introduces potential biases and limitations associated with data collection accuracy and consistency.

Suggestions:

  1. Increase Sample Size: Conducting further research with a larger sample size would provide more robust and reliable results. A larger sample size would improve the statistical power of the study and enhance the generalizability of the findings.

  2. Randomized Controlled Trial: Consider designing a randomized controlled trial (RCT) comparing the investigational treatment with standard treatments or a placebo. An RCT would provide a stronger level of evidence and enable better evaluation of the treatment's efficacy and safety.

  3. Multi-Center Study: Expand the study to include multiple medical centers and diverse patient populations to enhance the external validity and representativeness of the findings.

  4. Long-Term Follow-up: Extend the duration of the study and conduct long-term follow-up assessments to evaluate the treatment's durability, progression-free survival, and overall survival over an extended period.

  5. Include Comparative Analysis: Compare the investigational treatment with other existing treatment options to determine its comparative effectiveness and potential advantages in terms of response rates, survival outcomes, and safety profiles.

  6. Address Recruitment Challenges: Identify and address any barriers to patient recruitment to ensure the study reaches its targeted enrollment. This may involve collaboration with additional medical centers or implementing strategies to increase patient participation.

  7. Address Diversity and Inclusion: Make efforts to include a more diverse patient population by collaborating with medical centers serving diverse communities. This will help ensure the treatment's effectiveness across different racial and ethnic groups.If not in the scope of the manuscript, at least highlight the limitations. 

  8. Here are some suggestions for improving the discussion:

      • Condense the introduction by removing some of the historical context and details about individual studies. Focus on establishing the key challenges in treating relapsed/refractory MM and the need for effective options.

      • Break up the long paragraphs into shorter, more readable sections. Each new treatment/study could have its own paragraph for better flow.

      • Add a clear thesis statement upfront stating the purpose/objective of analyzing this particular study.

      • Strengthen the analysis and interpretation of the study results by directly comparing outcomes to stated goals/hypotheses and discussing implications/clinical significance.

      • Address potential limitations or weaknesses in the study design, sample size, patient population, etc. and how that may impact conclusions.

      • End with a brief conclusion summarizing what the study adds to the field and areas for future research, rather than ending abruptly after the results section.

      • Use section headings to better organize the different topics (e.g. Introduction, Treatments for RRMM, Study Objectives, Methods, Results, Discussion, Conclusion).

      • Proofread for typos, repetitive words, and clarity in wording/explanations throughout.

      • Consider adding a table to succinctly summarize key findings, treatment regimens, response rates etc. for easy comparison.

      • Check that all acronyms are defined on first use and references are formatted consistently.

      • This reviewer personally misses some thoughts out of the bo: 
      • Angiogenesis is an important process that tumors utilize to grow and spread. In multiple myeloma, new blood vessel formation supports the growth and survival of malignant plasma cells in the bone marrow microenvironment.

      • Drugs that inhibit angiogenesis may have potential benefits in the treatment of relapsed/refractory multiple myeloma. By cutting off the blood supply to tumor cells, anti-angiogenic drugs can impair myeloma growth and progression.

      • Combining anti-angiogenic drugs with other established anti-myeloma agents like proteasome inhibitors or immunomodulators could provide an effective dual approach - directly targeting tumor cells while also inhibiting their ability to stimulate new blood vessel formation. This may help overcome resistance seen with successive relapses.

      • Studies have shown that anti-angiogenic drugs such as bevacizumab have anti-myeloma activity when combined with proteasome inhibitors or other therapies. Exploring these types of combination regimens in relapsed MM may help improve outcomes.

      • Biomarkers related to angiogenesis could help identify patients most likely to respond to regimens that incorporate anti-angiogenic drugs. Measuring markers of angiogenesis could help guide treatment choices.

      • Future research should further explore incorporating anti-angiogenic therapies earlier in the treatment of MM, such as in induction or consolidation regimens before multiple relapses occur. This may achieve better long-term disease control.

    So in summary, targeting angiogenesis is a potential strategy to explore in the management of relapsed/refractory MM, particularly in appropriately selected combination regimens. Further research in this area may help advance treatment of this disease (please refer to 

      • PMID: 31936715 or to 
      • PMID: 29568363 and expand)
Comments on the Quality of English Language

Good.